# The Influence of the Depth of Cut in Single-Pass Grinding on the Microstructure and Properties of the C45 Steel Surface Layer

**DOI:** 10.3390/ma13051040

**Published:** 2020-02-26

**Authors:** Marek Szkodo, Karolina Chodnicka-Wszelak, Mariusz Deja, Alicja Stanisławska, Michał Bartmański

**Affiliations:** 1Department of Materials Engineering and Bonding, Faculty of Mechanical Engineering, Gdańsk University of Technology, G. Narutowicza Str. 11/12, 80-233 Gdańsk, Poland; mszkodo@pg.edu.pl (M.S.); michal.bartmanski@pg.edu.pl (M.B.); 2Department of Manufacturing and Production Engineering, Faculty of Mechanical Engineering, Gdańsk University of Technology, G. Narutowicza Str. 11/12, 80-233 Gdańsk, Poland; karolina.wszelak@pg.edu.pl; 3Department of Mechanics and Mechatronics, Faculty of Mechanical Engineering, Gdańsk University of Technology, G. Narutowicza Str. 11/12, 80-233 Gdańsk, Poland; alicja.stanislawska@pg.edu.pl

**Keywords:** single-pass surface grinding, medium carbon steel, indentation test, work hardening, surface layer

## Abstract

The paper contains the results of a metallographic examination and nanoindentation test conducted for the medium carbon structural steel with low content of Mn, Si, Cu, Cr, and Ni after its grinding to a depth ranging from 2 μm to 20 μm, at constant cutting speed (peripheral speed) of v_s_ = 25 ms^−1^ and constant feed rate of v_ft_ = 1 m/min. Applied grinding parameters did not cause the surface layer hardening, which could generate an unfavorable stress distribution. The increase in the surface hardness was obtained due to the work hardening effect. Microstructure, phase composition, and chemical composition of the grinded surface layer were examined using an X-ray diffractometer, light microscope, and scanning microscope equipped with X-ray energy-dispersive spectroscopy, respectively. Hardness on the grinded surface and on the cross-section was also determined. It was shown that the grinding of C45 steel causes work hardening of its surface layer without phase transformation. What is more, only grinding to a depth of 20 μm caused the formation of an oxide scale on the work-hardened surface layer. Nanoindentation test on the cross-section, at a short distance from the grinded surface, has shown that ferrite grains were more susceptible to work hardening than pearlite grains due to the creation of an equiaxed cellular microstructure, and that different dislocation substructure was created in the work-hardened surface layer after grinding to different depths.

## 1. Introduction

Grinding is a finishing abrasive treatment that allows receiving products with high dimensional accuracy and low surface roughness with a specific surface topography, which could be influenced even by the number of passes [1]. However, very often, the designers require not only a low roughness and high accuracy of the dimensions, but also the correspondingly high mechanical properties of the grinded surface layer. To meet the expectations of constructors, technologists plan heat treatment or thermo-chemical treatment of grinded products, i.e., volume or surface hardening, carburizing, or nitriding. Such additional technological processes, however, increase the production costs and extend the time of production. Moreover, heat treatment or thermo-chemical treatment requires the use of toxic coolants, which has a negative impact on the natural environment and on production workers who have direct contact with them [2]. Ball or slide burnishing can be used as alternative processes having no negative impact on the environment. In addition, during the burnishing process, a significant reduction in the surface roughness parameters and the surface texture of increased bearing capacity can be obtained [3,4,5,6]. Nevertheless, in recent years, the grinding technology combined with the simultaneous hardening of the surface layer has been intensively developing to eliminate these drawbacks [7,8]. During grinding, the surface layer of a workpiece heats up with a high rate [9,10,11], and it is subjected to considerable mechanical loads. The heat generated due to the friction between the workpiece and the grinding wheel increases the temperature of the processed workpiece. The increase of the temperature of the grinded surface layer of the workpiece above the Ac3 austenitization temperature (the temperature at which the transformation of ferrite to austenite is completed) and simultaneous quick cooling of the workpiece with the rate higher than the critical rate, results in surface hardening (Figure 1).

However, the introduction of this technology for the manufacture of components is faced with a number of difficulties. Grinding energy efficiency depends on the appropriate selection of cutting conditions, grinding wheel, workpiece material, and lubrication conditions [15,16]. There are difficulties with obtaining a repeatable result as it is difficult to achieve the assumed depth of hardening and to obtain the assumed hardness after grinding, which results in the necessity of applying the so-called softening grinding. In addition, the setting of grinding parameters thus as to obtain the temperature increase of the grinded surface, ensuring the transformation of pearlite and ferrite into austenite, results in a significant lowering of the yield strength of the heated layer. A significant reduction of the yield strength of the grinded layer in combination with the mechanical loads occurring during grinding hinders obtaining the assumed surface roughness. What is more, the stresses generated in the surface layer as a result of the grinding hardening have poor distribution, because under the hardened layer there are high tensile stresses while compressive stresses occur in the hardened layer. For example, Kruszyński and Wójcik concluded in their work [17] that, usually, residual stresses reach their maximum (tensile values) close to the surface on depths of 10–20 μm. A large gradient of residual stresses results in a reduction of the service life of the grinded elements. Rasmussen et al. [18] showed the wear of the rails after grinding hardening proceeded through cracks developing at the martensite/pearlite interface. Grinding induced surface tensile residual stresses were found to be the main factor causing the formation of micro-cracks on ground surfaces of 2304 duplex stainless steel during exposure with or without external loading [19]. Grinding with work-hardening of the surface layer can be an alternative method of grinding that will eliminate the disadvantages associated with grinding hardening. In order to ensure high hardness after grinding, the grinding parameters should be selected thus that a large fraction of the grinding energy falls on the plastic deformation of the material. At the same time, the thermal effects should be kept to a minimum. To avoid the effects of stress relief of the material, maximum temperatures in the contact zone should be as low as possible. This can be achieved by low cutting speeds in combination with low depths of cut. Such low temperatures, in addition to high plastics deformation of the material, results in a new advanced grinding process, which allows the combination of a shape generating grinding process and a mechanical subsurface strengthening process in a single-step machining operation [20]. The purpose of this work is to determine the influence of the grinding depth of C45 steel in a single-step grinding operation on susceptibility to the work hardening of pearlite and ferrite and determine the microstructure of hardened surface layer. The evaluation of the achieved work-hardening was done by nanoindentation test measurements.

## 2. Experimental Procedure

### 2.1. Preparation of Test Samples

Non-alloy medium carbon steel with a content of 0.47% C was used for the research. The chemical composition of the tested steel is presented in Table 1. The C45 steel specimen was cut from a rod with a diameter of 50 mm by wire electrical discharge machining (EDM) using an AccuteX AU-300IA machine (AccuteX Technologies, Taizhong, Taiwan). The height of the sample was 10 mm. After cutting, the sample was heat treated. The sample was annealed at 850 °C for 20 min and then cooled in ambient air. After heat treatment, the sample had a ferritic-pearlitic microstructure (Figure 2) with an average grain size of 20 μm. In the normalized state, the steel had the following mechanical properties: Tensile strength: 490 MPa and hardness: 167 HB.

### 2.2. Grinding the Sample

After heat treatment, the sample was grinded with various parameters, and then the hardness and topography of the grinded surfaces were determined. Two single grinding wheel transitions were made on both planes of the sample, each covering half the plane. In total, 4 grinding wheel transitions were made on both sides of the sample. Grinding was carried out on a grinding machine SPG 25 × 60 with a horizontal spindle axis and with computer numerical control (CNC) (FAS-Głowno, Głowno, Poland) (Figure 3). For machining, a Norton wheel (Saint-Gobain HPM Polska, Koło, Poland) with the designation 38A60LVS and dimensions (D × T × H) 250 × 25 × 76.2 mm was used. Each test was preceded by the conditioning of the grinding wheel on a single-grain diamond dresser with the following parameters: Depth at single pass: *a_e_* = 0.1 mm, number of cross-dressing transitions: 4, peripheral speed of the grinding wheel: *v_s_* = 23 m/s, lateral feed of the grinding wheel: *f_a_* = 0.2 mm/rev, number of spark-out passes: 2. The parameters have been set in the part program of the CNC controller. The grinding tests were carried out at a constant peripheral speed of *v_s_* = 25 ms^−1^ and constant feed rate of *v_ft_* = 1 m/min using flood cooling. A water-miscible, mineral oil-based cutting fluid Blasocut 2000 Universal (Blaser Swisslube AG, Hasle-Rüegsau, Switzerland) with 5% concentration was used as a coolant, and the flow rate was ${\dot{Q}}_{\mathrm{lub}}$ = 4.3 L/min. The variable parameter was the grinding depth set in subsequent tests according to the values: *a_e_* = 2; 8; 14; 20 μm. The chosen method of grinding was concurrent plunge grinding, with a fixed column.

### 2.3. Characterization

#### 2.3.1. Microstructure Analysis

The microstructure of the grinded surface layer was examined by X-ray diffractometer—XRD, with Cu Kα radiation λ = 0.15418 nm (Malvern, Worcestershire, UK), scanning electron microscope—SEM, JOEL JSM-7800F (Akishima, Tokyo, Japan) equipped with X-ray energy-dispersive spectroscopy EDS (Oxford Instruments Nanoanalysis, Abingdon, UK), and light microscope—LM Leica (Leica, Wetzlar, Germany), respectively.

#### 2.3.2. Nanoindentation Test

The hardness tests of the material were also performed after the grinding, using NanoTest Vantage nanoindenter (Micro Materials, Wrexham, UK). Hardness was measured, respectively, on the grinded surface and on the cross-section of the grinded surface. On the surface to be grinded, the hardness test was carried out at 3 different maximum loads of 50 mN, 100 mN, and 500 mN. For each load, 10 measurements were made, and the average arithmetic hardness was determined from them. On the cross-section, hardness tests were made along one section, parallel to the grinded surface at a distance of about 10 μm from the surface. This test was performed to determine the work hardening degree of the surface layer. A Berkovitch indenter was used for the hardness investigations. Hardness tests on the cross-section were performed with a constant load of indenter penetration equal to 20 mN. The maximum load during the nanoindentation test was chosen thus that the indenter’s impression could be entirely contained within the ferrite and pearlite grains. The time of force increase from the zero value was 20 s. Indentation contained one cycle with 5 s dwell at the maximum load. Before performing the nanoindentation test on the cross-section, the samples were grinded on abrasive papers with gradations from 60 to 2000. The samples were then polished mechanically using a diamond paste. In order to remove the Beilby layer after mechanical polishing, the sample was etched several times with nital (3.5% HNO_3_ solution in ethanol), and after etching, the etched layer was polished again.

## 3. Results and Discussion

### 3.1. Microstructure and Phase Composition of Grinded Surface Layer

Metallographic examinations using a SEM microscope showed different structures of the surface layer depending on the depth of grinding. Grinding at a depth of 2 μm did not change the pearlite into austenite. This means that the temperature in the surface layer did not exceed the Ac1 temperature at which austenite begins to form during heating. Because the heating rate of the surface layer during grinding was high, pearlite transformation initiates at higher temperatures of approximately 1050 K [21]. No oxides were formed on the grinded surface as the surface temperature was low, and the time was too short. Figure 4 shows the microstructure of the surface layer after grinding to a depth of 2 μm. The surface layer shows plastically deformed pearlite and ferrite grains and precipitation of cementite on the boundaries of ferrite grains. The biggest deformation was on the grinded surface and decreased as the distance from the surface increased. In ferrite grains, an equiaxed cellular structure was visible (see Figure 4b). In ferrite grains closer to the surface, the cells were larger in size. For example, the cell size in ferrite at a distance of 4 μm from the surface was 710 nm and at a distance of 10 μm was 430 nm on average. Figure 5 shows the ferrite grain model with the cellular structure. This model is closely related to the well-known model proposed by Mughrabi [22,23]. The Mughrabi model depicts crystal grain as the combination of material unit cells named by him “representative volume elements” (RVE’s). Unit cells contain a core region of low dislocation density and walls of high dislocation density.

The high density of dislocations in the walls makes them harder than the cores of cells with lower dislocation density. Accordingly, the grain can be represented as a two-phase material or multi-phase one, in general. Therefore, the model presented by Mughrabi in 1983 is the so-called composite model, and it is widely accepted by the scientific community.

During grinding, the highest temperature occurs in the contact zone of the tool and grinded material, and it decreases with the distance from the surface. Because the yield strength decreases with increasing temperature, dislocations can move more easily in the heated material. Dislocation mobility is the highest near the grinded surface and decreases with increasing distance from the surface. This, in turn, enables the dislocations that create cell walls to move to larger distances resulting in the formation of larger cells. Mughrabi has shown in his work [22] that the dislocations ensure compatibility of deformation by giving rise to long-range residual stresses, which superimposed on the applied stress, cause a redistribution of the stress on a local scale, thus permitting the simultaneous deformation of soft and hard regions. In this way, ferrite grains with cellular structure can be deformed during grinding. Pearlite grains with a lamellar microstructure consisting of cementite and ferrite can deform according to the same mechanism as the ferrite deformation mechanism described above. In the perlite, the role of hard walls to ensure compatibility between the deformation of hard cementite plates and soft ferrite lamellas provide cementite plates. In ferrite plates, no cellular structure was observed after grinding to a depth of 2 μm.

Grinding at a depth of 20 μm also does not transform the pearlite into austenite. However, in this case, there was a much larger share of thermal effects in the formation of the surface layer of the grinded material. Microscopic examination showed that there was an oxide layer on the plastically deformed material (see Figure 6a). The oxide scale has heterogeneous structures, as can be seen in Figure 6b. Figure 7 shows the XRD pattern and EDS spectrogram, and Table 2 presents the chemical composition of the oxide scale formed on the surface of C45 steel after its grinding to a depth of 20 μm.

During the grinding, the C45 steel was oxidized for a short time in the atmosphere and cooled quickly with air. The surface of the steel was covered with an oxide film constituted of magnetite (Fe_3_O_4_ of the inverse spinel structure), hematite (Fe_2_O_3_ of the corundum structure), and wustite or iron protoxide, which was always sub-stoichiometric (Fe_1−x_O of the standard structure NaCl) [25,26]. Figure 8 shows the iron/oxygen phase diagrams for the pressure of 1 atm. This diagram represents equilibrium conditions, and it never occurs in practice. As can be seen from Figure 8, only Fe_2_O_3_ and Fe_3_O_4_ exist in thermodynamic equilibrium. In turn, Fe_1−x_O is unstable below approximately 843 K. Since wustite is present in the oxide layer, it follows that the relatively fast cooling prevents the wustite from decomposing. Figure 9 shows a typical structure of the oxidized layer after iron oxidation at high temperatures [27].

During cooling, the wustite becomes unstable at temperatures above 843 K. Further cooling causes the “hypereutectoid” oxide to transform into oxygen-rich magnetite and ferrite. Because cooling takes place relatively slowly in the air, there is still enough time, and the temperature is high enough that the ferrite undergoes oxidation and transforms into wustite. That is why the precipitates of oxygen-rich magnetite are formed preferentially in the outer part of the wustite layer. During long term oxidation, the metal surface is also modified by various phenomena, such as internal oxidation or oxide penetration to the grain boundaries, and in the case of steel also decarburization. However, the oxidation kinetics of steel differs from the kinetics of pure iron oxidation. The oxidation kinetics of steel decreases when the concentration of the additional elements in alloy increases. Because the steel contained a small amount of alloying elements, the oxidation kinetics were high. This result was in agreement with the results of many studies carried out on the oxidation kinetics of iron alloys [27,28,29]. In a short time, an oxide layer with a thickness of about 2–3 μm was formed. As follows from the data presented in Table 2, the decrease in the iron concentration corresponds to the increase in the concentration of the oxygen and of the additional elements. For additional elements, their concentration in the oxide scale changes drastically compared to the concentration in steel (see Table 1 and Table 2).

Grinding to a depth of 8 μm and 14 μm causes only work hardening of the surface layer without the formation of the oxides scale. Figure 10c shows a work-hardened surface layer that has partially detached from the substrate. This effect could be caused either by exceeding the strength limit of the work-hardened surface layer or by thermal stresses because the surface layer during grinding heats up to higher temperatures than the material located at a greater distance from the grinded surface. The detached part of the work-hardened surface layer has a thickness of approximately 4 μm. The cellular structure was visible in strongly work-hardened ferrite grains (Figure 10a). In turn, in pearlite grains, ferrite and cementite tiles changed their direction due to strong deformation. A characteristic zig-zag structure was observed at the place of changing the direction of the tiles (Figure 10b). In place of strong deformation of pearlite grains, a cellular structure in ferrite lamellas was also formed (Figure 11). In the case of grinding to a depth of 14 μm, the detached part of the work-hardened surface layer had a greater thickness (even 10 μm) than in the case of grinding to a depth of 8 μm and, additionally, it was almost completely detached from the substrate, revealing a material with less work hardening degree. Figure 12 shows the microstructure of the surface layer after grinding to a depth of 14 μm. This figure shows the remains of the work-hardened and detached surface layer. There are numerous visible cracks in the work-hardened surface layer that indicate that the strength limit has been exceeded.

### 3.2. Hardness Measurements on the Grinded Surface

The measured hardness of the surface layer differs depending on the grinding depth, as well as the maximum load applied to the Berkovich indenter. Figure 13 shows the hardness measured on a C45 steel surface at various maximum loads of 50 mN, 100 mN, and 500 mN, respectively. Hardness measurements on the grinded surface were taken 10 times for each grinding depth and load, and the average arithmetic hardness along with the standard deviation was determined. As can be seen in Figure 13, the highest hardness of 7.53 GPa occurs for a surface layer grinded to a depth of 8 μm. During the nanoindentation test, the indenter loaded with a maximum force of 50 mN displaced to a contact depth of 476 nm ± 48 nm. In turn, at 100 mN and 500 mN loads, the contact depth was 718 nm ± 41 nm and 2455 nm ± 83 nm, respectively. As it results from graphs visible in Figure 13 for an analyzed grinding depth of 8 μm, at a contact depth of about 2.5 µm, the work hardening of the surface layer was already small, and its hardness was 3.47 GPa.

Work hardening of the surface layer grinded to a depth of 2 μm and 14 μm was very similar. This was due to the fact that although the work hardening of the grinded material to a depth of 14 μm was greater than when it was grinded to a depth of 2 μm, the part of the hardened surface layer detached from the substrate, revealing material with less work hardening. In turn, the hardness of the surface after grinding to a depth of 20 µm is due to the presence of an oxide film on its surface. Takeda et al. in work [31] gave hardness values for various iron oxides. According to them, the hardness of hematite, magnetite, and wustite was 6.70 GPa, 4.00 GPa, and 3.50 GPa, respectively. At a load of 50 mN, the indenter displaced to a depth of 570 nm, and the hardness of the oxidized layer was 6.28 GPa. This hardness was close to the hardness value for hematite given by Takeda. Lower hardness may result from hematite porosity. This observation was consistent with a recent study on oxidized high-speed steel [32]. At a 100 mN load, the indenter displaced to a depth of 853 nm, and the measured hardness was 5.6 GPa. Such a result of hardness measurement means that the oxidized layer consists of a mixture of hematite and magnetite at a distance of 853 nm from its surface.

### 3.3. Hardness Measurements on the Cross-Section of the Grinded Surface

Nanoindentation tests of the work-hardened surface layers were also carried out on a cross-section at a distance of about 10 µm from the cut surface (see Figure 14). Ten measurements were performed at a distance of 5 µm from each other, for all grinding depths. Pearlite and ferrite hardness was the arithmetic mean of results from as many measurements as the number of impressions after nanoindentation tests in the whole perlite or ferrite grains, respectively. If the impression area after the nanoindentation test covered both ferrite and pearlite grains, the results of the measurement were omitted in determining the hardness mean value. Due to that fact, the number of measurements of pearlite and ferrite hardness varied for different grinding depths, and the average arithmetic hardness along with the standard deviation was determined from them. Nanoindentation load-displacement curves provide a “mechanical fingerprint” of a material’s response to contact deformation. Figure 15 shows the loading and unloading curves obtained in the nanoindentation test for work-hardened ferrite grains, while Figure 16 shows the same curves for pearlite grains after grinding to different depths. As shown in Figure 15 and Figure 16, the indentation size effect (ISE) can be observed in indentation testing at different indenter displacements. A different slope of the loading curve can be observed up to about 50 nm for ferrite grains, while for pearlite grains, the slope change occurred at a smaller depth of about 40 nm. In addition, grinding to a smaller depth causes a change in the slope of the load curve for the smaller displacement of the indenter for both ferrite and pearlite grains. According to Nix and Gao [33], the ISE is directly related to geometrically necessary dislocations (GNDs), whose density is proportional to the inverse of the indentation depth. The density of GNDs is derived from the total line length of dislocation loops, necessary to form the shape of the indenter. The dislocations are geometrically necessary because they are introduced in the material in order to accommodate the shape of the indenter and thus provide the necessary lattice rotations. The formation of the GNDs occurs close to the surface, however, the storage volume was related to the area where the strain level was high enough to initiate plastic deformation. The shape and size of the plastic zone varied for different materials, and they were related, for example, by the crystal orientation [34]. The size of the plasticized zone also affected the tip rounding of the indenter and the change in the ratio of elastic and plastic deformation in the material given by a factor “modulus of elasticity”/”yield stress” (E/σy) due to the increased dislocation density at small scales [35]. For example, an increase in the E/σy ratio and in tip rounding would lead to a higher hardness. The discussed effects on plastic zone size could, again, cancel each other out. Thus, the shape of the load curve during the nanoindentation test depends both on the density of GNDs and the density of statistically stored dislocations (SSDs), created during grinding. As the indenter’s depth increases, SSDs have an increasing impact on the hardness measurement result because the density of GNDs is proportional to the inverse of the indentation depth. The above discussion explains the nanoindentation curves obtained for C45 steel grinded to different depths. For ferrite grains, ISE is visible up to an indenter displacement of 50 nm for grinding to a depth of 8 µm, 14 µm, and 20 µm, respectively. In turn, for grinding to a depth of 2 µm, this effect occurs for ferrite grains at a smaller indenter displacement of up to 40 nm. This may mean that grinding to greater depths generates a higher density of SSDs. The very high initial SSDs densities could lead to high repulsive forces between the SSDs and GNDs, which would increase the plastic zone size during indentation and, as a result, increase hardness [36].

During the indentation, the material around the contact area tends to deform downwards (sink-in) or upwards (pile-up) with respect to the indented surface. The occurrence of such “sink-in” and “pile-up” patterns is interpreted in terms of the work-hardening behaviour of the indented material [37,38].

According to these studies, the surface around indents tends to “sink in” against the indenter in cases where the indented sample is fully annealed and has a high work-hardening potential. In the case of highly work-hardened materials with only little reserves for further work-hardening or ones that generally have low work hardening potential, the surface around indents tends to “pile up”. As shown in Figure 14, ferrite and pearlite grains tend to “sink in” rather than “pile up”, despite the heavy deformation of the material. It rather seems that the nature of residual stresses occurring in the work-hardened material will determine the occurrence of “sink in” or “pile up”.

Another phenomenon that occurs during indentation is the so-called “pop in”. When shear stress underneath the indenter reaches a critical value given by the theoretical strength of the material, dislocation nucleation and spreading occurs. Immediately after “pop-in,” a very high density of GNDs is nucleated, and the GNDs dominate the deformation resistance of the material [36]. A “pop-in” event is then found in the load–displacement data. The critical load necessary to form a plastic impression is dependent on the density of SSDs and GNDs [36]:(1)$$F_{\mathrm{crit}}{\;=\;A}_{c}\cdot \mathrm{MC}\alpha \mathrm{Gb}\cdot \sqrt{\rho_{\mathrm{SSD}}{+\rho }_{\mathrm{GND}}}$$
where: Ac—surface contact, M—Taylor factor, α—factor dependent on dislocation substructure, C—constrain factor (transferring the complex stress state underneath the indenter in a uniaxial strain field, G—shear modulus, b—Burgers vector, ρ—density of SSDs or GNDs.

As shown in Figure 15 and Figure 16, “pop in” events are clearly visible in the entire load range for the ferrite and pearlite grains after grinding to a depth of 14 µm and 20 µm, respectively. This means that after grinding to a depth of 14 µm and 20 µm the density of SSDs was higher than after grinding to a depth of 2 µm and 8 µm or that the dislocation substructure varies greatly after grinding to different depths. Figure 17 shows an average hardness of ferrite and pearlite grains after grinding to different depths. The hardness measured on the cross-section is much higher than the hardness measured on the grinded surface. This may be due to the fact that, despite the occurrence of greater deformations on the surface, the higher temperature in these areas caused a softening effect on the material. In addition, grinding to a depth of 8 µm and 14 µm contributed to the cracking of the work-hardened surface layer, which also caused the relaxation of residual stress and reduced hardness. As can be seen from Figure 17, ferrite grains have a greater ability to work hardening than pearlite grains. Ferrite grains, after grinding to a depth of 2 µm, have the highest hardness of almost 11.5 GPa. Grinding to a greater depth causes the ferrite grains to have a much lower hardness ranging from 6.3 GPa to 7.5 GPa. A similar relationship between hardness and depth of grinding occurs in the case of pearlite grains, however, they show less ability to work hardening than ferrite grains. Macroscopic hardness is related to SSDs density, and it can be derived from the Taylor relations [39]:(2)$$H_{0}\;=\;\mathrm{MC}\alpha \mathrm{Gb}\cdot \sqrt{\rho_{\mathrm{SSD}}}$$

The high hardness of ferrite and pearlite grains after grinding to a depth of 2 µm indicates a higher density of SSDs. Therefore, “pop in” occurring in the indentation curves for pearlite and ferrite grains after grinding to a depth of 14 µm and 20 µm must result not from higher SSDs density but rather from a different dislocation substructure. Factor α presented in Equations (1) and (2) depends on the dislocation substructure. In the Taylor dislocation model, α is the empirical material constant between 0.1 and 0.5 [39]. From this, it follows that with the same density of SSDs, macroscopic hardness H0 may vary up to five times, depending on the dislocation substructure. When grinding to greater depths, the total heat flux intensity in the grinding area goes up, which leads to a change of the dislocation substructure and decrease of the hardness in the workpiece surface.

## 4. Conclusions

The medium carbon steel with a content of 0.47% C was single-pass grinded with various depths (2 µm, 8 µm, 14 µm, and 20 µm) and the microstructure and chemical composition of the grinded surface layer were examined subsequently by X-ray diffractometer, scanning electron microscope equipped with X-ray energy-dispersive spectroscopy (EDS), and light microscope (LM, Leica), respectively. The hardness on the surface and on the cross-section was determined. Furthermore, the results obtained in the nanoindentation test on a cross-section at a short distance from the grinded surface were also thoroughly analyzed. In addition, the microstructure of the oxidized surface layer after grinding to a depth of 20 µm was also discussed in detail. The obtained results allow forming the following conclusions:Grinding of C45 steel to a depth ranging from 2 μm to 20 μm at constant cutting speed (peripheral speed) of v_s_ = 25 ms^−1^ and constant feed rate v_ft_ = 1 m/min caused work hardening of the surface layer without phase transformation.The largest work hardening was shown by grinding to a depth of 2 μm, with greater hardness occurring at a distance of 10 μm from the grinded surface than on the cut surface. The high hardness after grinding to a depth of 2 μm resulted from the formation of a favorable equiaxial cellular structure in ferrite grains. In addition, grinding to a depth of 2 μm did not generate any cracks in the work-hardened surface layer.Grinding to a depth of 8 μm caused the work-hardening of the surface layer and its cracking due to exceeding the strength limit. In this case, the cellular structure was formed both in the ferrite grains and in the ferrite plates located in the pearlite. In the surface layer, the ferrite and cementite tiles are broken into a characteristic zig-zag, due to heavy loads.Grinding to a depth of 14 μm caused strong deformation of the grinded surface layer and its simultaneous detachment from the substrate with a much lower work hardening degree. In the ferrite grains and in the ferrite plates located in pearlite, the cellular structure is visible only directly at the surface, exposed due to the detachment of the hardened surface layer.The energy supplied to the material during grinding to a depth of 20 μm was used not only to work hardening of the surface layer but also to create an oxide scale that adheres well to the work-hardened substrate. This oxide scale consists of hematite, magnetite, and wustite.In all cases, ferrite grains were more susceptible to work hardening than pearlite grains, and after grinding to different depths, different dislocation substructure was created in the work-hardened surface layer.In further experiments, nanoindentation tests of the work-hardened surface layers will be carried out on a cross-section of the grinded surface at a varied distance from the cut surface. It will provide more detailed information about the properties of the hardened layer.

## Figures and Tables

**Figure 1 materials-13-01040-f001:**
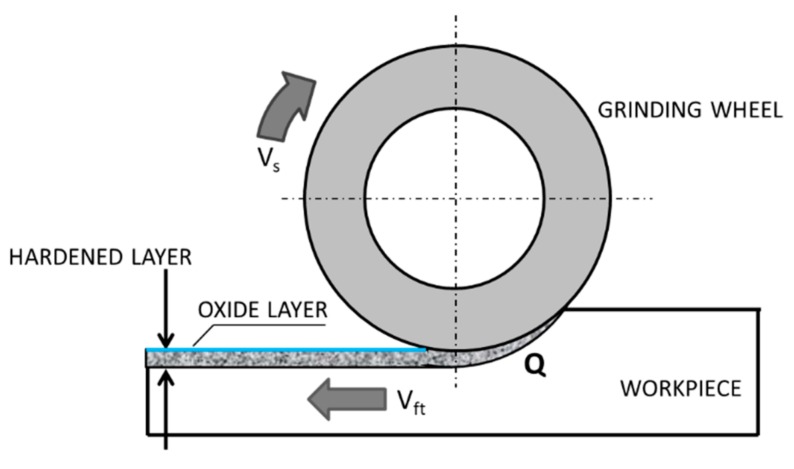
Grind-hardening process: v_s_—peripheral speed, v_ft_—feed rate, Q—heat rate entering the workpiece—based on references [12,13,14].

**Figure 2 materials-13-01040-f002:**
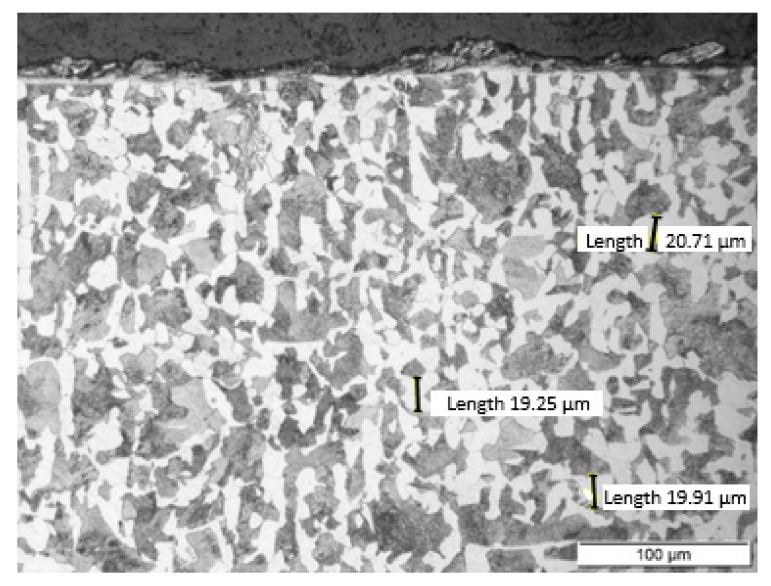
The ferritic-pearlitic microstructure of C45 steel after annealing. Sample etched with Nital. Photograph was taken using a light microscope LM Leica.

**Figure 3 materials-13-01040-f003:**
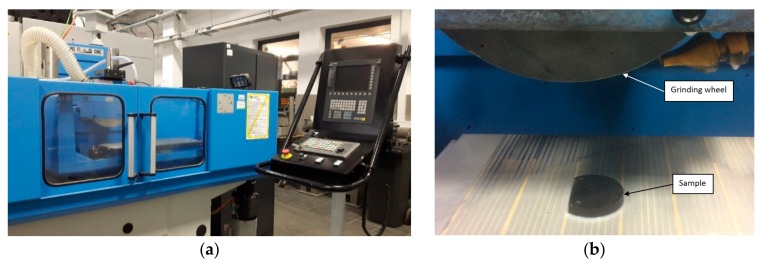
Horizontal-spindle (peripheral) surface grinder SPG 25 × 60: (**a**) General view, (**b**) working space with the sample location.

**Figure 4 materials-13-01040-f004:**
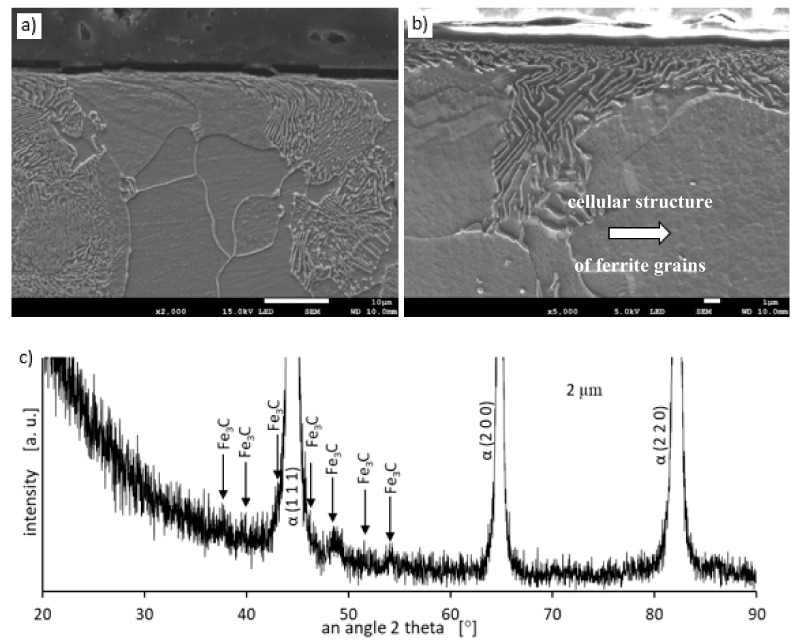
The microstructure of the surface layer after grinding to a depth of 2 μm: (**a**) Visible deformation of pearlite and ferrite grains in the direction of peripheral speed [24], (**b**) cellular structure of ferrite grains depicted by arrow, (**c**) X-ray diffractogram.

**Figure 5 materials-13-01040-f005:**
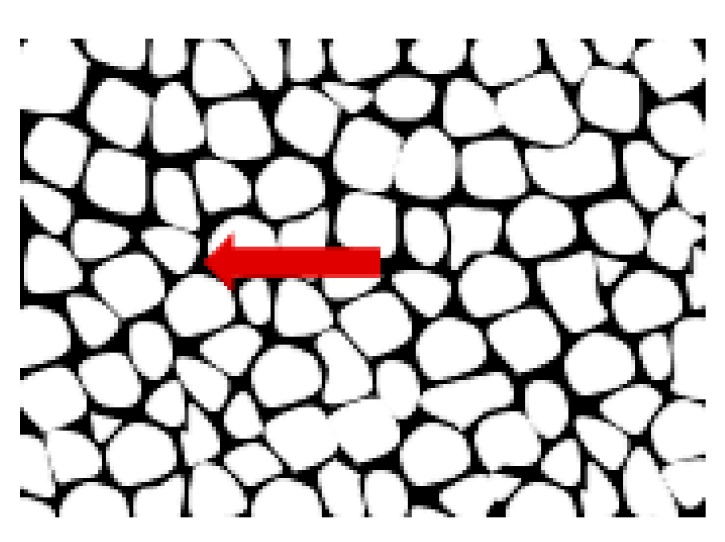
The model ferrite grain formed during grinding. The wall with high dislocation density is depicted by the arrow.

**Figure 6 materials-13-01040-f006:**
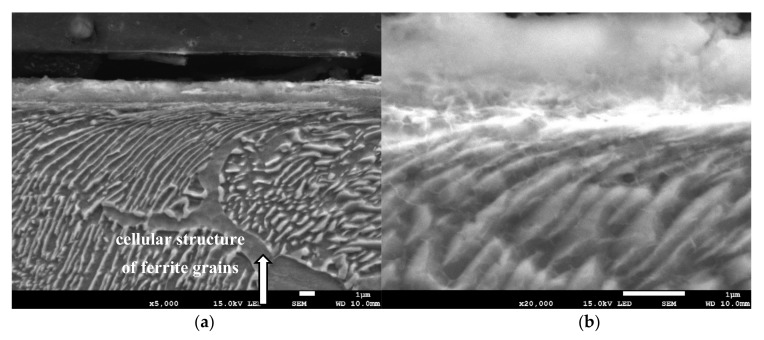
The microstructure of the surface layer after grinding to a depth of 20 μm: (**a**) Deformation of pearlite and ferrite grains in the direction of peripheral speed—cellular structure of ferrite grain is depicted by an arrow, (**b**) scale with heterogeneous structures on the deformed layer.

**Figure 7 materials-13-01040-f007:**
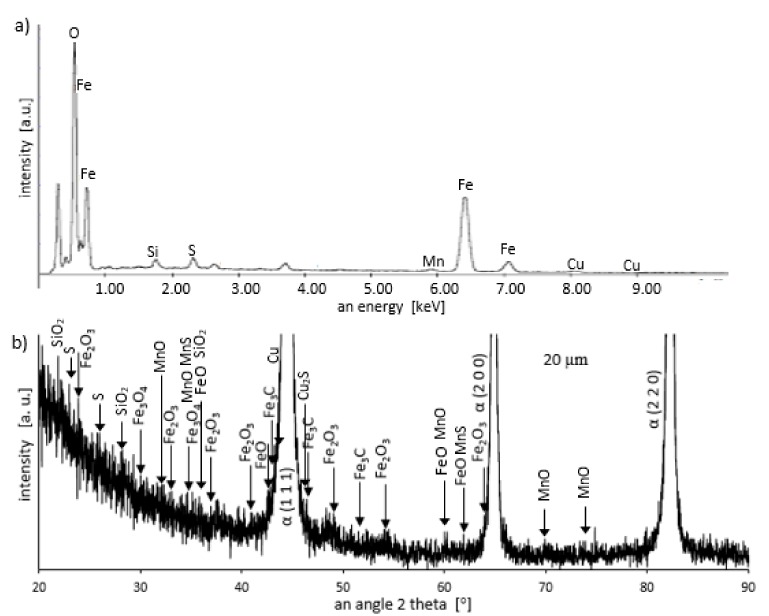
Energy dispersive spectroscopy (EDS) spectrogram (**a**) and X-ray diffractometer (XRD) pattern (**b**) for a sample grinded to a depth of 20 μm.

**Figure 8 materials-13-01040-f008:**
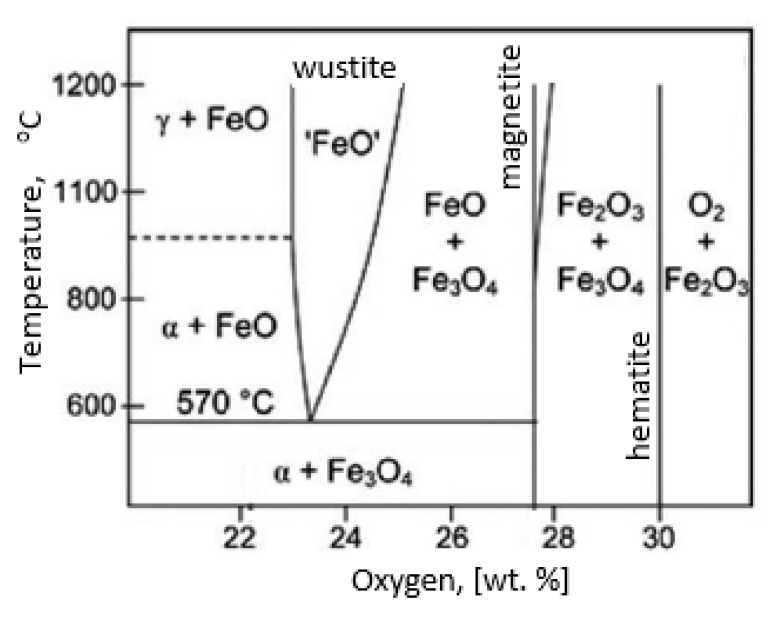
Iron/oxygen phase diagrams for the pressure of 1 atm—based on reference [30].

**Figure 9 materials-13-01040-f009:**
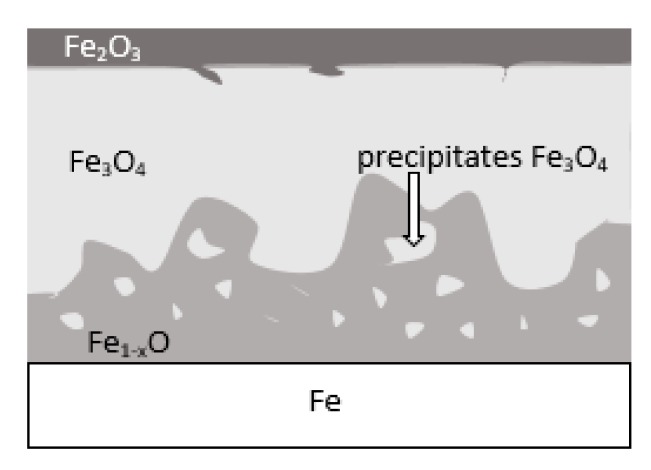
Schematic representation of the iron surface after oxidation in air at high temperature—based on reference [27].

**Figure 10 materials-13-01040-f010:**
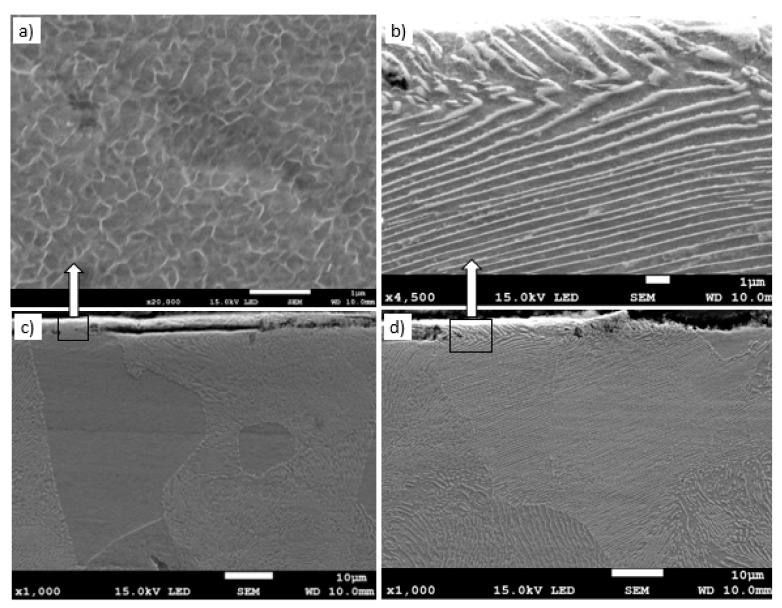
The microstructure of the surface layer of the C45 steel after grinding to a depth of 8 μm with deformation of ferrite (**a**), (**c**) and pearlite (**b**), (**d**) grains in the work-hardened surface layer. Part of the work-hardened surface layer is detached from the substrate (c).

**Figure 11 materials-13-01040-f011:**
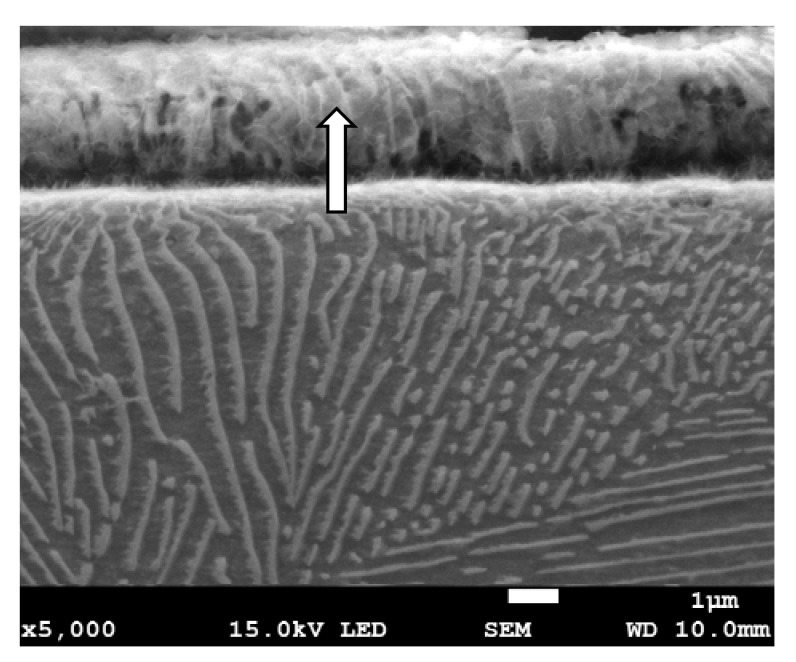
The microstructure of the surface layer of the C45 steel after grinding to a depth of 8 μm with deformation and partially detached of pearlite grain. In the detached part of pearlite, cellular structure in ferrite plates is visible (depicted by arrow).

**Figure 12 materials-13-01040-f012:**
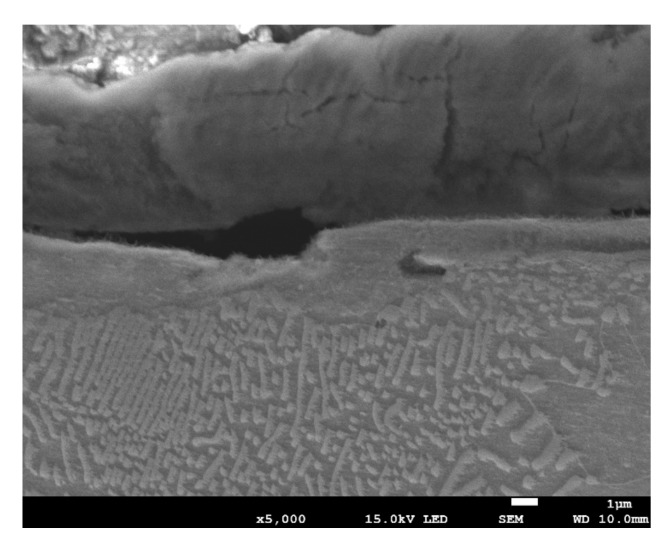
The microstructure of the surface layer of the C45 steel after grinding to a depth of 14 μm.

**Figure 13 materials-13-01040-f013:**
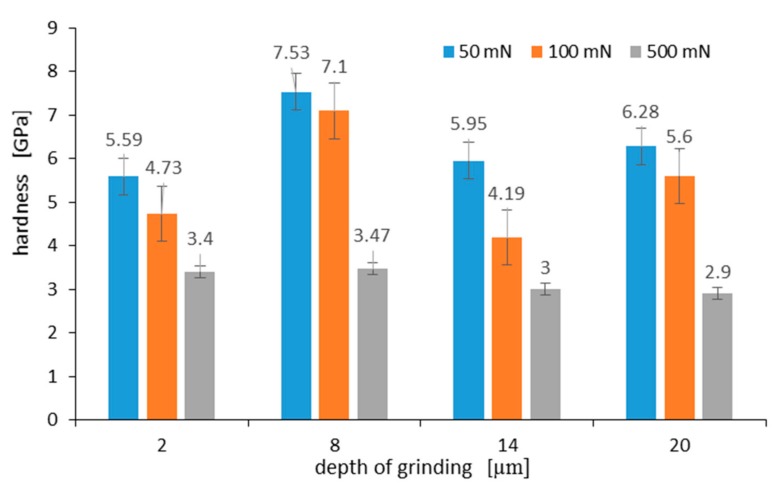
Hardness measured on C45 steel surface at various maximum loads after grinding to a depth of: 2 μm, 8 μm, 14 μm, and 20 μm. Error bars show ± standard deviation.

**Figure 14 materials-13-01040-f014:**
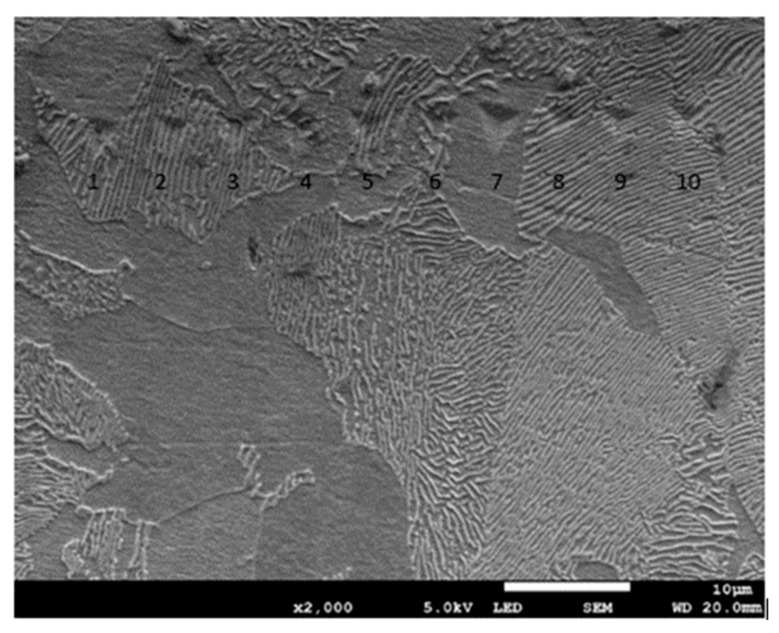
Micrograph showing impressions after the nanoindentation test for C45 steel grinded to a depth of 2 μm.

**Figure 15 materials-13-01040-f015:**
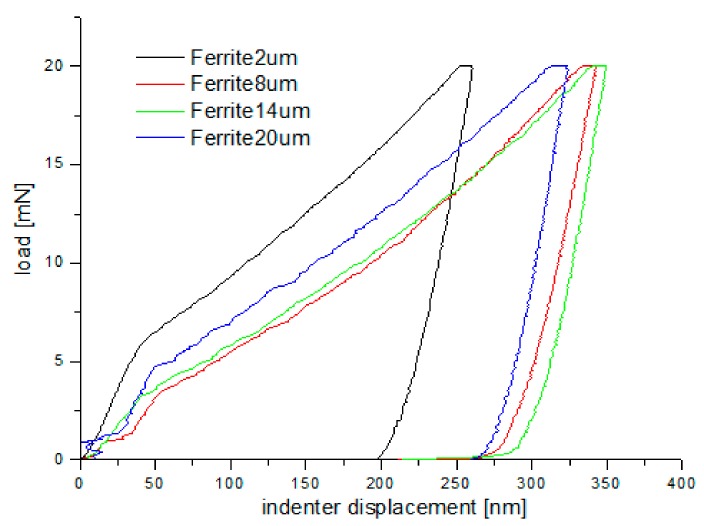
Loading–unloading curves obtained on the cross-section, in the nanoindentation test, for ferrite grains after grinding to different depths.

**Figure 16 materials-13-01040-f016:**
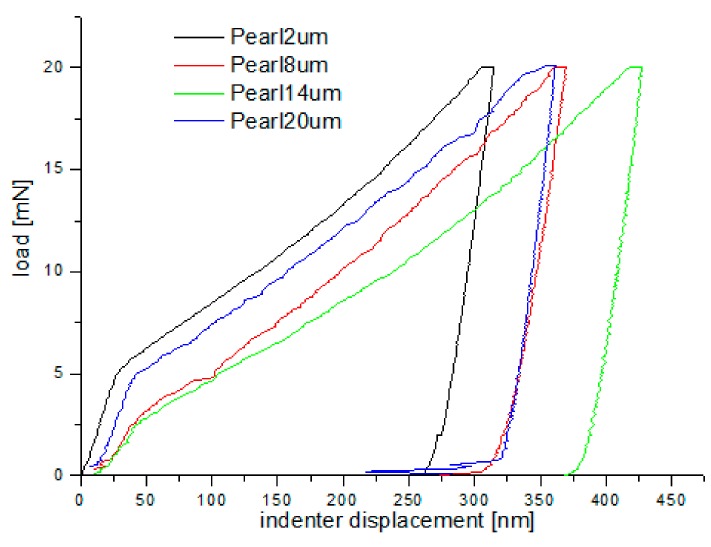
Loading–unloading curves obtained on the cross-section, in the nanoindentation test, for pearlite grains after grinding to different depths.

**Figure 17 materials-13-01040-f017:**
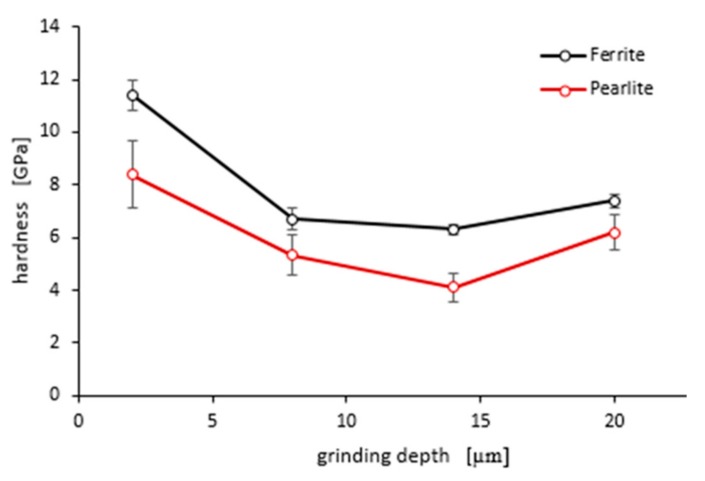
An average hardness of ferrite and pearlite grains after grinding to different depths. Error bars show ± standard deviation.

**Table 1 materials-13-01040-t001:** Chemical composition of tested steel.

Chemical Composition wt.%							
C	Mn	Si	P	S	Cu	Cr	Ni
0.47	0.65	0.27	0.030	0.025	0.25	0.17	0.26

**Table 2 materials-13-01040-t002:** Chemical composition of the oxide scale formed on the surface of C45 steel after its grinding to a depth of 20 μm.

Element	wt.%	at.%
O	32.82	62.05
Si	0.98	1.05
S	1.52	1.43
Mn	1.47	0.81
Cu	2.34	1.12
Fe	60.87	33.54
Total	100	100

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
