# Peer review of "The Influence of the Depth of Cut in Single-Pass Grinding on the Microstructure and Properties of the C45 Steel Surface Layer"

_materials, 2020, doi:10.3390/ma13051040_

Round 1

Reviewer 1 Report

This paper presents the influence of single-pass surface grinding parameters on the microstructure and properties of the C45 steel surface layer, which would be of interest to a broad audience in relevant field. I have read this manuscript with a great pleasure and the authors have done a great job in presenting an interesting and well-structured manuscript.

Could authors explain in detail why they chose C25 steel and what is the significant of present study with other commonly used steel.  Could authors explain in detail about the statistical analysis methods used in this paper.  Also, sample size for nanoindentation is not enough for  Authors have mentioned manufacturability is one of the major concern for Grind-hardening process, could authors comment on how their work would help in addressing this problem.

Author Response

Dear Reviewer,

our responses to your comments are presented in the attachment.

With kind regards,

Authors

Reviewer 2 Report

The paper is well presented and provides the reader with sufficient information to understand the issue the authors are dealing with. I acknowledge the effort of the authors to reinvent the analysis of grinding, taking into account that this process has been previously extensively studied and referenced. I have some comments that should be cleared out for publication:

The abstract presents correctly the issue you are tackling. However, it should also be explicit about the technological impact of your results (it is actually explained in the introduction, just reflect it in the abstract as well). As grinding is a finishing process, and you highlight the fact that it leaves tensile residual stress on the surface, you should also refer to other alternative processes that could solve this problematic, such as ball burnishing. You should therefore name this fact in the introduction (around Line 44) and include references such as the following in the new version for its publication: 10.1016/j.surfcoat.2018.05.061 & 10.1016/S0924-0136(03)00750-7 Line 143. I have doubts about how hardness has been measured. If you polish, then etch the surfce with Nital to remove the Beilby layer, you cannot know whether you are actually excessively disolving the ferrite, and that could affect the indentation resolts. Could you clear this out?

Author Response

(The authors gave the same response as above.)

Reviewer 3 Report

An interesting and valuable paper

Not particularly innovative but of industrial relevance

Author Response

Dear Reviewer, thank you for your comments. As you wrote, we hope that our work will help to conduct this kind of method in industrial practice and in further research. We would like to inform you that we have introduced some modifications according to the comments of other Reviewers and that revisions have been made to the maximum extent. 

Authors

Reviewer 4 Report

Language and structure:

The paper is well structured and the language is correct. The only suggestion in this sense would be to substitute “grinded” when referring to a surface machined by the grinding process and use “groud” instead.

Extension:

The extension of Section 3.1 is too large. Authors should try to reduce its length (pages 6 and 9 include too long explanations). Reducing its length will help readers to better understand the discussion. Authors could alternatively try to divide it in subsections for improved readability.

Figures:

Authors should improve the aspect of Figure 5 and Figure 8, their quality is below the average quality of the rest of the figures. Figure 13 must be improved. The same scale must be used for all graphs. Authors should also consider including all the results in one graph. If authors show results for each maximum load together, the effect of grinding depth will be more visible.

Experimental:

Hardness tests of on the cross section must be better explained. Figure 14 shows 10 indentations at the same depth, but authors do not clearly explain how many tests have been carried out in total. Authors should also give information about dispersion presented by these results. Authors must perform more indentation tests at different depths. Current indentations have been carried out at 10µm depth, but the size of the indentation marks is small enough to perform indentation tests at 5 µm and 20µm for example. This would provide a lot of information about the extension and properties of the hardened layer.

Author Response

(The authors gave the same response as above.)

Reviewer 5 Report

The authors have studied the influence of single-pass surface grinding parameters on the microstructure and properties of C45 steel surface layer. The following are the reviewer comments:

  1. Constant peripheral speed (Vs) of 25m/s and constant feed rate (Vft) of 1m/min were employed. Please justify the parameters selection. The influence of the parameters may only be studied if authors vary atleast one of the parameters such as Vs or Vft and reprot the temperature build up on the workpiece that in turn affect the properties.  
  2. Also, what is the temperature of the workpiece after grinding? The properties (microstructural and hardness) of the steel workpiece relies on the temperature build up due to cutting and this should be reported. Relationship between cutting parameters and temperature buildup would be important. Further, Coolant properties and flow rate should be reported.

Author Response

(The authors gave the same response as above.)
